# The Impact of Treadmill Training on Tissue Integrity, Axon Growth, and Astrocyte Modulation

**DOI:** 10.3390/ijms25073772

**Published:** 2024-03-28

**Authors:** Tatyana Ageeva, Davran Sabirov, Albert Sufianov, Eldar Davletshin, Elizaveta Plotnikova, Rezeda Shigapova, Galina Sufianova, Anna Timofeeva, Yuri Chelyshev, Albert Rizvanov, Yana Mukhamedshina

**Affiliations:** 1OpenLab Gene and Cell Technology, Institute of Fundamental Medicine and Biology, Kazan Federal University, 420008 Kazan, Russia; 2Department of Neurosurgery, Sechenov First Moscow State Medical University of the Ministry of Health of the Russian Federation (Sechenov University), 119991 Moscow, Russia; 3Research and Educational Institute of Neurosurgery, Peoples’ Friendship University of Russia (RUDN), 117198 Moscow, Russia; 4Department of Pharmacology, Tyumen State Medical University, 625023 Tyumen, Russia; 5Department of Histology, Cytology and Embryology, Kazan State Medical University, 420012 Kazan, Russia; 6Division of Medical and Biological Sciences, Tatarstan Academy of Sciences, 420111 Kazan, Russia

**Keywords:** spinal cord injury, treadmill training, neuron, astrocytes, axon growth

## Abstract

Spinal cord injury (SCI) presents a complex challenge in neurorehabilitation, demanding innovative therapeutic strategies to facilitate functional recovery. This study investigates the effects of treadmill training on SCI recovery, emphasizing motor function enhancement, neural tissue preservation, and axonal growth. Our research, conducted on a rat model, demonstrates that controlled treadmill exercises significantly improve motor functions post-SCI, as evidenced by improved scores on the Basso, Beattie, and Bresnahan (BBB) locomotor rating scale and enhanced electromyography readings. Notably, the training facilitates the preservation of spinal cord tissue, effectively reducing secondary damage and promoting the maintenance of neural fibers in the injured area. A key finding is the significant stimulation of axonal growth around the injury epicenter in trained rats, marked by increased growth-associated protein 43 (GAP43) expression. Despite these advancements, the study notes a limited impact of treadmill training on motoneuron adaptation and highlights minimal changes in the astrocyte and neuron–glial antigen 2 (NG2) response. This suggests that, while treadmill training is instrumental in functional improvements post-SCI, its influence on certain neural cell types and glial populations is constrained.

## 1. Introduction

The restoration of motor function is a critical priority for individuals suffering from traumatic spinal cord injury (SCI), regardless of its severity, chronicity, or the age at onset [1,2]. Developing interventions that maximize improvement within the limited time and resources available for rehabilitation is crucial for optimizing outcomes related to motor functions [3]. One key approach to enhancing sensory and motor functions post-SCI involves stimulating neural pathway activation to promote regeneration and growth of descending and ascending pathways involved in spinal, supraspinal, and/or afferent inputs [4,5,6]. In clinical practice, a comprehensive suite of rehabilitation activities, including treadmill training with body weight support, has been employed as an effective tool for activating these pathways [7,8,9]. However, the mechanisms underlying functional recovery following SCI remain unclear and necessitate further exploration, including through animal models.

Significant restoration of motor function in paralyzed hind limbs has been reported in animal models of SCI following treadmill training [10,11,12,13]. Studies in rats have demonstrated that such training not only significantly prevents secondary damage [14] but also enhances neuroplasticity by increasing or restoring the expression of neurotrophins post-SCI [15]. In various experiments, motor exercises have been shown to amplify brain-derived neurotrophic factor (BDNF) mRNA expression and protein levels in both intact and injured spinal cords, notably in the ventral horn [16,17,18,19]. Treadmill training has also been observed to reduce oxidative stress and increase BDNF in the brain neurons in a rat model of Parkinson’s disease [20]. Such training prevents atrophy of spinal motoneurons in SCI [21], inhibits neuronal apoptosis by activating cyclic adenosine monophosphate/protein kinase A (cAMP/PKA), protein kinase B (Akt), and glycogen synthase kinase 3beta (GSK3β) pathways in rat brains post-ischemia [22], and promotes the sprouting of lateral axon branches in SCI [23], as well as axonal regeneration in peripheral nerve damage [24].

It is important to note that motor training can directly affect the state of neurons and their extensions, as well as indirectly through other cellular components of the spinal cord. In the study by Ying et al., 2022 [25], it was shown that water treadmill training facilitates axon growth, associated with the Nogo receptor/Rho-associated protein kinase (NGR/RhoA/ROCK) signaling pathway, by inhibiting the activation of astrocytes. A significant reduction in pro-inflammatory factors induced by SCI, such as interleukin-1β, -6 (IL-1β, IL-6) and TNFα, was reported following the training [26].

Despite the extensive body of research focused on the effects of motor rehabilitation after SCI, existing data on molecular and cellular changes in neural tissue in response to exercise remain contradictory. This variability is largely due to differences in training methodologies, including the timing of initiation, duration, and the specificity of animal support during exercise. There are a noted lack of studies that simultaneously focus on the use of mild spinal cord injury and enforced plantar placement of animal limbs during training sessions. Such an approach could potentially shed light on optimizing conditions for the recovery of motor functions and neuroplasticity, as well as on the mechanisms underlying the efficacy of rehabilitative training.

In this study, we devoted considerable attention to the stimulating influence of motor training on the recovery of motor function, the preservation of neural tissue, and the study of neuronal plasticity: the possibilities of modulating the phenotype of neural cells, as well as the expression and cellular composition of the glial scar.

## 2. Results

### 2.1. Training Treadmill Stimulates the Motor Function Recovery

*BBB.* During the first week following surgery, rats from both groups experienced the effects of SCI, including lack of sensory response to paw stimulation, lower-limb paraplegia, and urinary incontinence. During the early recovery period, at 6–9 dpi, both groups exhibited gradual improvement in motor function, indicating the onset of recovery (Figure 1a). From 11 dpi, the Trained group showed significant improvement in BBB scores (p=0.0064), indicating more pronounced recovery of motor functions compared with the Untrained group. Improvements were noted in the ability to support the body, motor activity in the joints of the hind limbs, and reduced clearance. Overall, comparative analysis of the groups revealed statistically significant differences between the Trained and Untrained groups from day 11 to day 28. The BBB score in the group with controlled rehabilitation was approximately 1.5 times higher than in the Untrained group throughout the experiment. The highest level of motor activity was recorded at 28 dpi in the Trained group, reaching [10.92 ± 2.81] points on the BBB scale, while, in the Untrained group, this indicator was [6.61 ± 3.01]. The study results emphasize that controlled treadmill training significantly contributes to the improvement of motor functions in rats following SCI.

*EMG.* To further assess the understanding of the effects of rehabilitation training, an electromyographic analysis was conducted. This analysis aimed to evaluate the impact of rehabilitation procedures on nerve conductivity and the functional state of muscles in rats, providing us with a deeper understanding of the recovery processes. At 28 dpi, we observed a significant decrease in the amplitude of the M-response in the experimental groups, with significant differences between the Trained and Intact control groups [Trained 25.47 ± 3.79 µV vs. Intact 40.1 ± 7.14 µV, p<0.05] (Figure 1b). During the same period, the amplitude of the H-response also decreased 2.5–3.7-fold (p<0.05) in the Untrained [3.02 ± 1.29 µV] and Trained [2.04 ± 1.44 µV] groups compared with the group of intact animals [7.69 ± 3.63 µV] (Figure 1b’). We did not observe intergroup differences (Untrained vs. Trained) in the ratio of H-wave amplitude to M-wave amplitude at 28 dpi.

SEPs in intact rats at the lumbar level were represented by P1-N1-P2 peaks (Figure 1c). Lumbar SEPs were found in 100% of intact animals. The average amplitude of peak N1 in intact rats was 1.89 ± 0.09 µV, with a latency of 15.7 ± 1.53 ms. At 28 dpi, we registered lumbar SEPs only in 27% and 41% of cases in the Untrained and Trained groups, respectively. There was a sharp decrease in the average amplitude of peak N1 in both the Untrained [0.99 ± 0.16 µV] and Trained [1.34 ± 1.71 µV] groups compared with intact animals. The recorded potential from scalp electrodes appeared as a P1–N1 complex, in which the amplitude and latency of N1 were assessed. Scalp SEPs were found in 75% of intact animals. The average amplitude of peak N1 in intact rats was 29.03 ± 7.52 µV, with a latency of 2.22 ± 0.14 ms. On the 28th day of the experiment, we recorded scalp SEPs in 100% and 92% of cases in the Untrainedand Trained groups, respectively. There was also a decrease in the average amplitude of peak N1 in both the Untrained [18.31 ± 7.69 µV] and Trained [22.03 ± 8.13 µV] groups compared with intact animals.

MEPs from the gastrocnemius muscle were registered in 100% of intact animals, with an amplitude of 15.95 ± 3.84 mV and a latency of 5.08 ± 0.95 ms. At 28 dpi, MEPs were recorded in 63% and 33% of the Untrained and Trained groups, respectively. The average amplitude of recorded MEPs in the Trained group was 0.34 ± 0.23 mV, with a latency of 9.48 ± 5.74 ms, and in the Untrained group—0.86 ± 0.29 mV and 12.56 ± 3.93 ms without intergroup differences.

### 2.2. Treadmill Training Maintained Tissue Integrity

To elucidate the stimulating mechanisms of motor function recovery in rats, we conducted a morphological analysis of the injured spinal cord using histological staining of longitudinal sections in the contusion injury area, 5 mm rostral to 2.5 mm caudal from the point of impact (Figure 1d,e). The selected area of the spinal cord longitudinal section was measured in mm^2^, and the area fraction was expressed in percentage of pixels in a row (Figure 1f,f’). To demonstrate the percentage of preserved neural tissue, intact spinal cord tissue accounts for 99.47%. The recorded value in the intact sample is associated with visualization of the central canal and minor tissue artifacts.

At 28 dpi, in the samples of the injured Untrained vs. Trained groups, tissue preservation on average decreased to 60.39% vs. 70.81%. After inverting the threshold digital image, all existing pathological cavities were highlighted (Figure 1e), but only those larger than 200 µm^2^/15 pixels were measured. Pathological cavities on average accounted for 39.61% vs. 29.19% of the total area of the longitudinal section in the Untrained vs. Trained rat groups, respectively. We also analyzed differences in tissue preservation in the dorsal and ventral directions. It was found that, in the injured groups, the number and area of pathological cavities in the dorsal part of the spinal cord were significantly higher than in the ventral part (Untrained: 44.61% (dorsal) vs. 30.85% (ventral); Trained: 25.37% (dorsal) vs. 16.66% (ventral)), which is an expected result since the impact force and, consequently, the primary SCI were most intense in the area of the corticospinal tract and dorsal roots. The distribution of pathological cavities in the dorso–ventral and rostro–caudal directions was presented in the form of three-dimensional reconstructions of contusion SCI lesions, projected based on serial longitudinal sections.

Analysis showed a significant correlation (r = −0.967, p<0.05) between the BBB score at 28 dpi and the size of pathological cavities and preservation of spinal cord tissue in the Trained group.

### 2.3. Treadmill Training Stimulates Axonal Growth

At 28 dpi, the expression of the GAP43/CSPG4 combination was analyzed on longitudinal sections (Figure 1g). GAP43 was expressed in the area 6–8 mm caudal to the injury in rats in both the Untrained and Trained groups. GAP43 expression around the lesion occupied 16.71 ± 4.27% of the total row area in the Trained group, which was higher (p<0.05) than in the Untrained group at 11.01 ± 3.69% (Figure 1h). These longitudinal sections were simultaneously incubated in antibodies against the CSPG4 marker of NG2 glia, which has an inhibitory effect on axonal growth. The expression of CSPG4 in the same area of longitudinal sections decreased 3.9-fold in the Trained group compared with the Untrained group.

### 2.4. Limited Impact of Treadmill Training on Motoneurons

Behavioral assessment, EMG, and tissue preservation indicated the restoration of functional connections in the spinal cord under rehabilitation conditions. These results prompted us to evaluate the phenotype of thoracic neurons at 6–8 mm caudal to the epicenter of the injury, which interact during the activation of lumbar circuits during training. In examining combinations of OPN/ChAt (Figure 2a–d) and PARV/bTubIII (Figure 2e,f), no significant differences were found in the VH of the spinal cord between the Untrained and Trained groups. There were also no significant differences observed in the expression of the chat gene (Figure 2h). Additionally, we analyzed the potential enhancement of synaptogenesis. However, no significant differences in the expression of the *syp* gene were found between the groups (Figure 2g).

The localization of ACAN in PARV^+^- and NeuN^+^/ChAt^+^-neurons in the VH of the spinal cord was established. Most NeuN^+^/ChAt^+^-neurons were found to be expressing ACAN, as confirmed by the identical dynamics of these two populations at 28 dpi. In the Trained group, there was a significant increase in the number of ACAN^+^/PARV^+^-neurons compared with the Untrained group [4.5 ± 2.5 vs. 10.5 ± 2.5] at 28 dpi.

### 2.5. Treadmill Training Promoted Anti-Inflammatory Reactive Astrocytes

The destructive contribution in SCI is observed from reactive astrocytes. Using a combination of (GFAP/ALDH1L1), we analyzed the population of astrocytes expressing ALDH1L1 and/or GFAP in the white and gray matter zones in the injury area at 28 dpi (Figure 3a). We identified the presence of at least 3 different phenotypes: GFAP^+^/ALDH1L1^−−^-, GFAP^−−^/ALDH1L1^+^- , and GFAP^+^/ALDH1L1^+^-astrocytes. Significant differences were found in the number of GFAP^+^-astrocytes (phenotypes GFAP^+^/ALDH1L1^−^ + GFAP^+^/ALDH1L1^+^) in the CST and VH of the spinal cord at 6–8 mm caudal to the injury site (Figure 3b). In the VH, the number of GFAP^+^-cells was twice as high (p<0.01) in the Trained group compared with the injured spinal cord without rehabilitation. In the CST area, this indicator was 40% higher (p<0.01) in the Trained group compared with the Untrained group. Additionally, the number of GFAP^+^/ALDH1L1^+^-cells in the VH significantly decreased (p<0.05) 1.3-fold after treadmill training compared with the Untrained group (Figure 3c). The population of GFAP^−^/ALDH1L1^+^-astrocytes did not show significant differences in the white and gray matter areas of the spinal cord at 6-8 mm caudal to the injury site (Figure 3d). The increase in the number of GFAP^+^-astrocytes was confirmed by the analysis of mRNA GFAP levels, which showed a more than 3.4-fold increase (p<0.05) at 6–8 mm caudal to the injury site in the Trained group compared with the Untrained group (Figure 3e). Also, an increase in the number of S100A10^+^-astrocytes after SCI and training was shown, but no significant differences were registered between the studied groups for this criterion (Figure 3f). This cell line was studied only in the VH at 6–8 mm caudal to the injury site.

### 2.6. Stability of NG2 Glia and Limited Astrocytic Response to Treadmill Training Post-SCI

We also conducted an assessment of changes in the expression of CSPG4 proteoglycan by astrocytes, as a separate and unique line of astrocytes after SCI, analyzing the combination of GFAP/CSPG4 in the VH at 6–8 mm caudal to the injury site (Figure 4a). The assessment of the average fluorescence intensity of CSPG4 proteoglycan in the VH of the spinal cord showed no significant differences between the Trained and Untrained groups at 28 dpi (Figure 4b). An analysis of the expression of CSPG4 proteoglycan in GFAP^+^-astrocytes was also conducted (the level of colocalization of CSPG4/GFAP was determined) using the Pearson’s coefficient, which was 0.05747 (with a 10% to 90% range of 0.04659 to 0.07108) in the Trained group and 0.05494 (with a 10% to 90% range of 0.05406 to 0.06583) in the Untrained group (Figure 4c). No statistically significant differences were found between the experimental groups regarding CSPG4-producing GFAP^+^-astrocytes. Also, no significant differences were recorded between groups in the expression of the *ng2/cspg4* gene at 6–8 mm caudal to the injury epicenter.

## 3. Discussion

Treadmill training is increasingly recognized as a standard and non-invasive method for treating the aftermath of SCI, particularly becoming a principal therapeutic tool in conditions of mild spinal cord damage. Our study reinforces the notion that controlled treadmill exercises significantly contribute to the improvement of motor functions in rats following SCI. This is consistent with findings from other other research groups [27,28,29], which have highlighted the benefits of motor training in post-SCI recovery, particularly in terms of neurological assessment using the BBB scale. Our findings demonstrate that treadmill training can address dysfunction in conducting pathways resulting from SCI in animals, thereby aiding the restoration of movement in hind limbs. The simulated SCI in our study led to the loss of descending inputs into voluntary and automated motor schemes of the spinal cord, resulting in partial paralysis below the injury level. Our study confirms the positive impact of treadmill treatment on the recovery of neural conductivity and muscle functions after SCI. Despite the reduced presence of SEPs in the Trained group compared with intact animals, they were still observed more frequently than in the Untrained group, which may indicate an improvement in sensory functions. In the Trained group, MEPs were recorded more often than in the Untrained group, suggesting an enhancement of motor functions and a potential improvement in the state of motor neurons. Presumably, the excitability of motor neurons around the injury site increases over time due to a series of adaptations in spinal cord circuits [30,31], which is associated with improved motor function [32,33].

Post-injured SC in rats often leads to neural tissue necrosis and cavity formation, progressively shrinking the spinal cord, especially in later injury phases [34,35]. Treadmill training has been found to improve the condition of post-traumatic tissue in the injury area, demonstrating a restraining effect on secondary damage and fiber preservation. Our data align with the findings of Ying et al., 2020, which demonstrate a reduction in blood–spinal cord barrier (BSCB) permeability and decreased structural damage to tissues under water treadmill training conditions [32]. However, the presence of residual/preserved fibers in the white matter alone is insufficient to initiate voluntary muscle contractions. This underscores the importance of growth and regeneration of damaged axons for full functional recovery after SCI. Wang et al., 2023, reported that treadmill training alone facilitates growth/regeneration of axons in the CST [36]. Our results indicate a significant increase in the expression and fluorescence intensity of GAP43 protein, as well as the area of GAP43 immunopositive axons following treadmill training. This aligns with other studies showing that similar training promotes neurite growth in post-injured SC, including the increased expression of GAP43 and neurofilament 200 (NF200) proteins, and GAP43 mRNA [25]. Sabatier et al. (2008) also demonstrated elongation of afferent axons in the peripheral nerve following motor load [23]. Nevertheless, axonal regeneration alone cannot lead to adequate recovery of motor function unless these regenerated axons are appropriately connected. Treadmill exercises contribute to the improvement of postsynaptic density, evident from heightened levels of postsynaptic density protein 95 (PSD95) and synaptosomal-associated protein 25 (SNAP25) [37]. Additionally, recent studies indicate an augmentation in synaptic transmission and increased neuronal activity, and enhanced myelination of axons (increased intensity of myelin basic protein) through the activation of mTOR signaling in layer 5 pyramidal neurons, as indicated by increased phosphorylation of ribosomal protein S6 [37,38]. Higher expression of vGlut1^+^-motoneurons was reported in rats trained on a treadmill after spinal cord transection [39] and in mice with contusive SCI [40]. Treadmill exercises also increased synaptic coverage of vGlut1^+^ terminals on motoneurons after sciatic nerve transection [41]. The increase in vGlut1^+^ profiles suggests an increase in presynaptic terminals and glutamatergic synapses. However, it is to be noted that an increase in the number of presynaptic/postsynaptic elements is not proof of the presence of synapses. In our study, we encountered insufficient and contradictory results, which do not confirm the impact of treadmill training on synaptogenesis. The absence of significant differences in syp gene expression emphasizes the need for a deeper analysis of synaptic proteins and other molecules involved in the process of synaptogenesis.

It has been demonstrated that physical exercises can reverse or prevent disruptions in GABAergic and glycinergic regulation, increasing the expression of glutamate decarboxylase 67 (GAD67), glycine receptor (GlyR), and gamma-aminobutyric acid subtype A (GABAA) in the lower thoracic Th10 and lumbar L2 segments [42], via BDNF-TrkB-dependent pathways in SCI [43]. It has been shown that interneurons with increased expression of GAD67, GlyR, and GABAA play a role in locomotion, as they can modify the pace of the stepping cycle, and their reduction post-SCI may lead to prolonged activation of spinal motoneurons [44,45]. These motoneurons innervate flexor muscles, disrupting the stereotypical flexion–extension phases of contraction during locomotion [46]. The lack of effect of treadmill training on the subpopulation of ChAt^+^-, OPN^+^-, and PARV^+^-motoneurons and the expression of the *chat* gene in our study might seem counterintuitive, especially in the context of results from other studies. For instance, Lin et al., 2023, demonstrated a significant increase in the number of ChAT^+^ -motoneurons at the C4 level (rostral to the injury site) [38].

We studied the motoneurons in the VH of IX lamina, which are of interest in terms of the microenvironment analysis. Specifically, the study of astrocyte profiles, as close participants in the interaction with motoneurons, plays a key role in the response to SCI and secondary damage [47]. GFAP is one of the key markers used for identifying astrocytes in the CNS and is an indicator of astrocytic activity and pathology [48,49]. Active training increases the level of GFAP [50,51]. Physical exercises are capable of increasing the density of GFAP^+^-astrocytes and cellular expression in the CA1 area of the rat hippocampus [52]. The increased immunoreactivity of GFAP in astrocytes is often associated with their hypertrophy, but this may exaggerate the actual extent of changes. Assessing the increase in the number of GFAP^+^-cells as a sign of their local recruitment or proliferation might reflect physiological adaptive plasticity, rather than just a response to pathological stimuli [53]. In our study, the increase of GFAP^+^-cells in the VH and CST, without changes in the morphology of astrocytes and their processes towards reactivity under motor load (Trained), can be interpreted as a positive effect. These data may indicate an increase in trophic support for motoneurons in the gray matter and support for regenerating descending axons in the white matter.

In a number of studies focused on the examination of cellular phenotypes during astrogliosis, the importance of employing additional astrocytic markers for a more detailed understanding of the role of GFAP is emphasized [53,54]. The use of the ALDH1L1 marker expands the ability to detect astrocytes that cannot be solely identified through GFAP staining. In our samples with injury, as expected, most astrocytes were also positive for GFAP. Consequently, two primary groups of astrocytes were identified: GFAP^−^/ALDH1L1^+^ (non-reactive astrocytes) and GFAP^+^/ALDH1L1^+^ (reactive astrocytes). The reduction in the number of GFAP^+^/ALDH1L1^+^-astrocytes in the group with controlled training in our study also supports the hypothesis of a positive effect of physical exercise on the state of the glial scar. Recent studies have demonstrated that the microenvironment of the glial scar gradually transforms, inhibiting regeneration as the scar matures, and the function of astrocytes progressively shifts from stimulating to inhibiting axonal regeneration [55,56]. Limiting secondary damage caused by inflammation may be a specific function of astrocytes, precursors of ependymal cells (one of the three astrocyte lineages identified in SCI [55,56,57,58]), including S100A10^+^- reactive astrocytes. An increase in these cells after training indicates the induction of morphological changes and proliferation of astrocytes, aimed at forming neuroprotective phenotypes. Similar results were obtained in the study by Ying et al., 2022, where it was found that training on a water treadmill promotes the development of S100A10^+^-reactive astrocytes [25]. This same study also demonstrated a reduction in pro-inflammatory astrocytes (the second line of astrocytes, known as resident cells [55,56,57,58]) expressing the C3 protein. The existing discrepancy between the observed levels of C3 mRNA in our work and the C3 protein in Ying et al., 2022 [25] may be due to differential regulation of translation, post-translational modifications, protein half-life, and antibody epitope availability.

Another representative marker gene of astrogliosis is CSPG4, often associated with so-called NG2 cells (the third lineage of astrocytes identified in SCI) [59]. It has been shown that NG2 glia in the spinal cord after injury can generate a small portion (5%) of new astrocytes in both gray and white matter [60]. In our study, we demonstrated that the colocalization of GFAP and CSPG4 markers is very weak (Pearson coefficient less than 0.1) in both groups, with an insignificant reduction in the training group. This corresponds to the findings of other studies on the lack of significant contribution of astrocytes to the change in CSPG4 expression [60,61] following SCI, and likely indicates that such GFAP expression is merely a temporary activation of the GFAP gene. Overall, the NG2 glia population remained stable after training, with no significant differences in CSPG4 expression between the two groups.

## 4. Materials and Methods

### 4.1. Animals

We purchased 50 female Wistar rats (12–16 weeks old, weighing 200–250 g), all sourced from BioNurse STEZAR, Russia. The rats were housed in standard breeding cages, with free access to food and water in a controlled environment of constant temperature (23 ± 1 °C), 12-h light–dark cycles, and 60–65% humidity. The animals were randomly assigned into three groups: Intact (n = 10, no surgical interventions performed), Untrained (n = 20, rats underwent the standard SCI model, details below), and Trained (n = 20, similar to the Untrained group, with the addition of treadmill training starting on day 8 post-injury). The choice of 50 rats for this study was based on a power analysis to ensure adequate statistical power for detecting meaningful differences across treatment conditions, factoring in expected dropout rates and variability in response to SCI and rehabilitation.

### 4.2. Rat SCI Model Construction

Animals were anesthetized using isoflurane (1.3%) and Zoletil (20 mg/kg). Under aseptic conditions, the simulated surgery group exposed the dura mater of the T8–T9 segment, with a width of 0.5 cm. The spinal cord was exposed under the pia mater, and injury was inflicted using the Impact One Stereotaxic Impactor-CCI (Leica Microsystems, Durham, NC, USA) with an impact force of 1.5 m/s. The signs of a successful confirmation of the model are the appearance of unsteady swing and tail flick reflex in rats. Post-surgery, animals received daily gentamicin (5 mg/kg) for seven days and underwent manual bladder emptying until the return of the voiding reflex.

### 4.3. Motor Training Protocol

Motor training on the Treadmill for Mice and Rats (IITC Life Science, Woodland Hills, CA, USA) commenced on day 8 post-injury and was conducted for 5 days each week. Training sessions lasted for 20 min, twice a day, with a 2-h interval. The manual enforcement of weight-supported plantar foot placement on the treadmill was maintained until the animal could perform these actions independently [62]. The treadmill speed was adjusted based on the animal’s functional status, which was assessed using the Basso, Beattie, and Bresnahan (BBB) locomotor rating scale, more details of which are described below. The initial treadmill speed was set at 6 cm/s (3.6 m/min). Upon reaching a BBB score of 7, indicating extensive movements in all three joints, the speed was increased to 21 cm/s (12.6 m/min). Rehabilitation lasted for 3 weeks.

### 4.4. Behavioral Test

The BBB scoring method, which employs a scale ranging from 0 to 21, was used to assess the recovery of voluntary movement, where higher scores indicate better movement recovery [63]. Animals were acclimatized in an open field for 3 days pre-surgery. Each rat was individually placed in an open field at 7 days post-injury (dpi) and observed for three minutes by two researchers conducting the study blindly. The procedure was repeated every 2–3 days thereafter.

### 4.5. Electromyography

Stimulated electromyography was conducted using an 8-channel electroneuromyograph Neuro-MVP-8 (Neurosoft, Ivanovo, Russia) at 28 dpi. Somatosensory evoked potentials (SEPs) and motor evoked potentials (MEPs), and M- and H-waves of the tibialis anterior and gastrocnemius muscles were recorded in response to sciatic nerve stimulation. Monopolar needle electrodes with length 22 mm and diameter 0.40 mm (#MN4022D15SRU, Neurosoft, Ivanovo, Russia) served as active (inserted into the muscle mid-belly) and reference (at the Achilles tendon) electrodes. Electrical stimulation used single rectangular pulses (0.2 ms). For motor evoked potentials, transcranial stimulation was applied with needle electrodes penetrating subcutaneously into the skull, using pulses of 0.04 to 0.1 ms and intensities between 20 and 500 V; more details were described in previous studies [64,65].

### 4.6. Morphometric Analysis

Spinal cord segments (5 mm rostral to 2.5 mm caudal) centering on the injury epicenter were used to assess neural tissue preservation and pathological cavities. After embedding in Tissue-Tek O.C.T. Compound, 20 µm longitudinal sections were prepared using a Microm HM 560 cryostat. Seventeen slides (obtained from the dorsal to the ventral part of the spinal cord), each containing 3 sections, were selected for each animal, with the epicenter of the lesion and the ependymal canal used as references. Sections were stained with hematoxylin and eosin and examined with an APERIOCS2 light scanning microscope. Morphometric analysis was conducted using NIH ImageJ software (https://imagej.nih.gov/ij/, accessed on 15 April 2023). Digital images were converted to 8-bit grayscale, and tissue and cavities were quantified based on brightness levels. Huang’s automatic thresholding method distinguished stained tissue from the background. Area and brightness statistics were calculated, and cavities exceeding 200 µm^2^ were identified and measured. Representative three-dimensional reconstructions were created from a selected set of slices in the required order. These models were subsequently visualized in a three-dimensional projection utilizing Blender (https://www.blender.org, accessed on 1 September 2023).

### 4.7. Immunofluorescent Staining

After the last behavioral test and EMG at 28 dpi, rats were anesthetized by intramuscular injection of tiletamine/zolazepam (30 mg/kg, Zoletil, Virbac Lab, Carros, France) and xylazine (6 mg/kg, Xylanit, Nita-Farm, Saratov, Russia) and perfused transcardially with 0.01 M phosphate-buffered saline (PBS; pH 7.4), followed by 4% paraformaldehyde in PBS. The extracted spinal cords were sectioned into 20 µm slices and prepared for staining. For the general analysis of glial scarring, transverse sections were used, taken from the Th9 segment located 6–8 mm caudal to the injury epicenter. In addition, for the analysis of GAP43 expression, longitudinal sections were specifically prepared. The sections underwent a series of washing and blocking steps, followed by incubation with primary antibodies for antigen identification (see Table 1). Fluorescently labeled secondary antibodies were then applied to visualize antibody binding, with cell nuclei counterstained using 4′,6-diamidino-2-phenylindole (DAPI). Detailed imaging was carried out using an LSM 700 confocal microscope (Carl Zeiss Microscopy GmbH, Rostock, Germany), focusing particularly on the glial scar analysis. To assess areas associated with potential glial scarring and glial fibrillary acidic protein/aldehyde dehydrogenase 1 family member L1 (GFAP, ALDH1L1), the following regions of the spinal cord were selected, in line with our previous studies [66,67]: ventral horn (VH), corticospinal tract (CST) in the dorsal columns, ventral funiculi (VF), central canal (CC), and the dorsal root entry zone (DREZ). We studied the motoneurons in the area of the VH of IX lamina. Immunopositive cells within a 0.05 mm^2^ area of each section were quantified across six sections at 0.5 µm intervals using NIH ImageJ software (Bethesda, ML, USA, https://imagej.nih.gov/ij/, accessed on 15 April 2023). Controls were stained with secondary antibodies alone to confirm the specificity of the staining protocol. The colocalization of neurons and cells was investigated and quantified using Zen Black software (Carl Zeiss Microscopy GmbH, Rostock, Germany), accessed on 1 September 2020). The Pearson’s coefficient was analyzed using the JaCoP plugin NIH ImageJ software (Bethesda, MD, USA, https://imagej.nih.gov/ij/, accessed on 15 April 2023). This method implies the use of the following estimates to characterize the degree of colocalization, taking into account the values of the Pearson correlation coefficient: 0.1—very weak, 0.2—weak, 0.3—less weak, 0.4—less than moderate, 0.5—moderate, 0.6—more than moderate, 0.7—less than strong, 0.8—strong, 0.9—very strong.

### 4.8. RT-qPCR

At 28 dpi, animals were anesthetized and spinal cord segments were promptly harvested and preserved at −70 °C. The preserved tissue was later used for RT-PCR to assess gene expression relevant to glial and fibrotic scars, myelination, microglia, and neurons. Cell samples were thawed for RNA extraction using TRIzol and processed through centrifugation, chloroform, and isopropanol treatment, and then precipitated and washed with ethanol. The RNA was air-dried and resuspended in water. RNA concentration was quantified using a NanoDrop device. The RNA was primed and reverse-transcribed into cDNA using RevertAid Reverse Transcriptase. This cDNA served as a template for RT-PCR on a CFX96 ThermoCycler (BioRad, Hercules, CA, USA). The reaction mix included a specialized buffer, Taq DNA polymerase, deoxynucleotide triphosphates (dNTPs), and gene-specific primers and probes listed in Table 2, with denoting glyceraldehyde 3-phosphate dehydrogenase (GAPDH) rRNA serving as an endogenous control. The amplification followed the TaqMan protocol, with data from triplicate independent experiments. The relative expression levels of genes were analyzed using the $2^{-\Delta \Delta \mathrm{Ct}}$ method.

### 4.9. Statistical Software and Data Analysis Methods

Statistical analysis of results was conducted using Origin 10.0 SR0 (OriginLab Corporation, Northampton, MA, USA) and R 3.6.3 (R Foundation for Statistical Computing, Vienna, Austria) software. Repeated measurements were averaged to derive a single mean per animal; these means were subsequently combined to determine the overall group average. In cases of non-normal distribution, the median was used instead of the mean to represent central tendency. Statistical significance of differences was determined using Kruskal–Wallis analysis with subsequent Dunn’s post hoc test for RT-PCR results, as well as Student’s *t*-test and Mann–Whitney test for BBB, EMG, and immunofluorescence histochemistry results. The Pearson correlation was carried out to quantify the correlation between BBB scores and histological data of pathological cavities. For all statistical data, a significance level of less than 0.05 (p<0.05) was adopted.

## 5. Conclusions

The study demonstrates that dosed treadmill training offers limited benefits for normal walking recovery in hind limbs post-mild contusion injury and supports spinal tissue integrity post-injury, alongside promoting axonal growth. However, its lack of effect on motoneurons and minimal impact on astrocytes and NG2 glia highlight the complexity of SCI recovery. These findings underscore the importance of adopting integrated rehabilitation strategies in clinical practice to fully enhance recovery potential following spinal cord injuries.

## Figures and Tables

**Figure 1 ijms-25-03772-f001:**
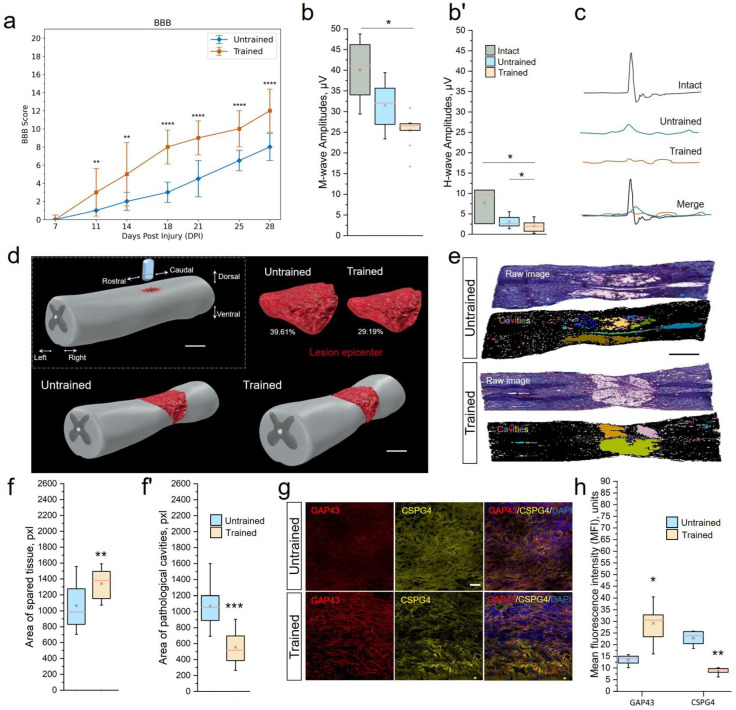
Treadmill training restores motor function, maintains tissue integrity, and supports axonal growth at 28 dpi (3 weeks of motor training). (**a**) Locomotor recovery measured by BBB rating scores. Maximum score—21. The electrophysiology results of M-wave amplitudes (**b**), H-wave amplitudes (**b’**) and somatosensory evoked potentials (SEPs) (**c**). (**d**) Three-dimensional reconstructions of the lesions. Scale 1 mm. (**e**) Representative images of quantitative assessment of neural tissue integrity in longitudinal sections (5 mm rostral to 2.5 mm caudal) of Untrained and Trained groups. Digital RGB color images (raw) of the spinal cord after hematoxylin–eosin staining. Binary inverted images after conversion to 8-bit with the application of Huang’s automatic threshold method and overlay masks. Identified cavities are colored, larger than 200 µm^2^. Scale 1 mm. Area quantification of spared tissue (**f**) and pathological cavities (**f’**). (**g**) Expression of growth-associated protein 43 (GAP43) and chondroitin sulfate proteoglycan 4 (CSPG4) in the spinal cord, longitudinal section, 6–8 mm caudally. Confocal microscopy. Scale 20 µm. (**h**) Quantitative graph of the GAP43 and CSPG4 areas. For BBB test (**a**): Mann–Whitney test with Bonferroni correction; for EMG (**b**,**b’**): Kruskal–Wallis test with Dunn’s multiple comparison; for tissue integrity analyze (**f**,**f’**) and assessment of axonal growth (**h**): Mann–Whitney test. For all (**a**,**b**,**b’**,**f**,**f’**,**h**) values expressed as medians (minimum, maximum, and first and third quartiles (25th and 75th percentiles)): Mann–Whitney test. For all (**a**,**b**,**b’**,**f**,**f’**,**h**) values expressed as medians (minimum, maximum, and first and third quartiles (25th and 75th percentiles)), x—mean, +—outliers, n=20 rat/Untrained and Trained, n=10 rat/Intact group for (**a**,**b**,**b’**) and n=10 rat/group for **f**, **f’**,**h**; * p<0.05; ** p<0.01; *** p<0.001; **** p<0.0001.

**Figure 2 ijms-25-03772-f002:**
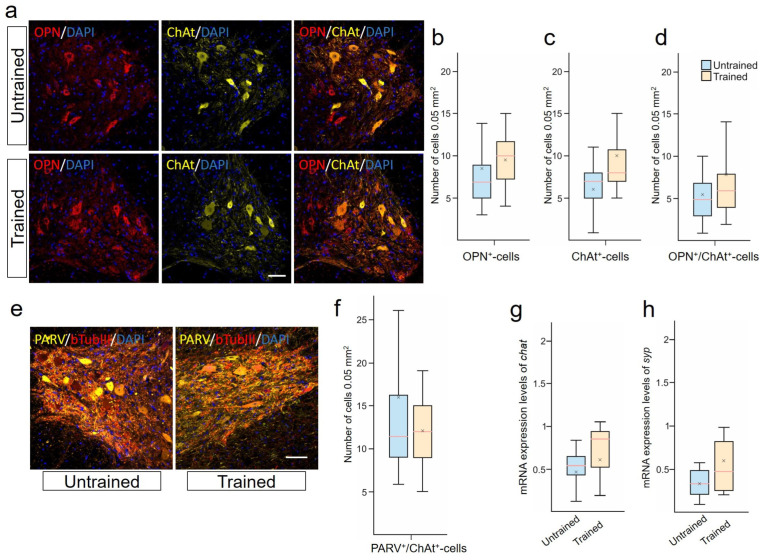
Limited impact of treadmill training on motoneurons. (**a**) Confocal microscopy of osteopontin (OPN) (red) and choline acetyltransferase (ChAt)^+^ (yellow) neurons in the ventral horns (VHs) of the spinal cord, 6–8 mm caudal to the epicenter of injury at 28 days post-injury (dpi). Nuclei are stained with DAPI (blue). Scale 100 µm. OPN^+^- (**b**), ChAT^+^- (**c**) and OPN^+^/ChAT^+^-cells (**d**) quantification at 28 dpi. (**e**) Confocal microscopy of bTubIII^+^ (red) and PARV^+^ (yellow) neurons in the ventral horns of the spinal cord, 6-8 mm caudal to the epicenter of injury. Nuclei are stained with DAPI (blue). Scale 100 µm. (**f**) PARV^+^/bTubIII^+^-cells quantification. qRT-PCR was performed to determine the expression of chat (**g**) and syp (**h**) at 28 dpi. The data are expressed as $2^{-\Delta \Delta \mathrm{Ct}}$. For all (**b**–**d**,**f**–**h**): Mann–Whitney test. Values are expressed as medians (minimum, maximum, and first and third quartiles (25th and 75th percentiles)), x—mean, n=5 rat/group, no significant differences between groups.

**Figure 3 ijms-25-03772-f003:**
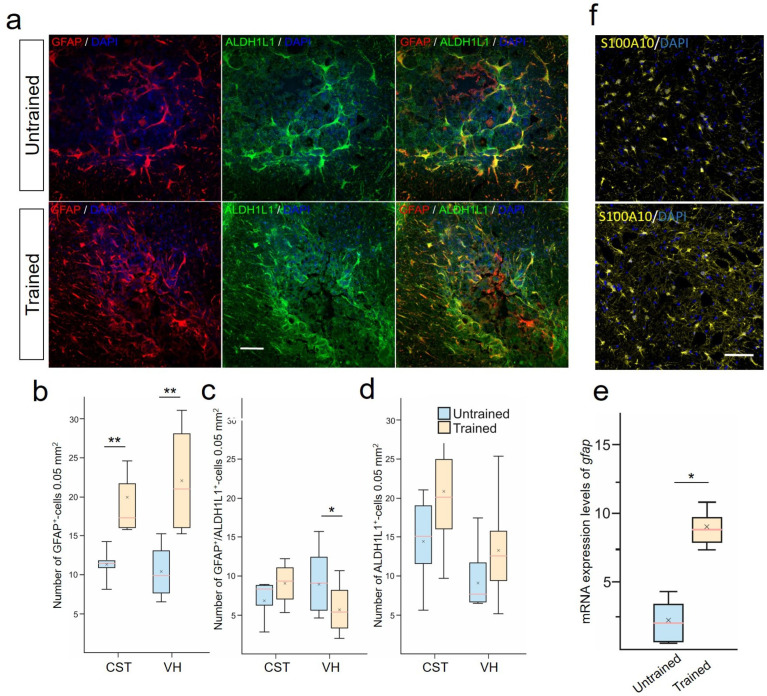
Treadmill training promoted the development of anti-inflammatory reactive astrocytes. (**a**) Combined staining for aldehyde dehydrogenase 1 family member L1 (ALDH1L1) (green) and glial fibrillary acidic protein (GFAP) (red). ALDH1L1 marks both the cell bodies and extensive processes of astrocytes in the CST in spinal cord, 6–8 mm caudal to the epicenter of injury. GFAP labels the intermediate filament cytoskeleton of astrocytes. Confocal microscopy. Scale 100 µm. Quantitative graphs of different populations of astrocytes: GFAP^+^-cells (includes GFAP^+^/ALDH1L1^−^ and GFAP^+^/ALDH1L1^+^-astrocytes) (**b**), only GFAP^+^/ALDH1L1^+^-cells (**c**), and stable population of ALDH1L1^+^-(or GFAP^−^/ALDH1L1^+^)-astrocytes (**d**). qRT-PCR was performed to determine the expression of gfap (**e**) at 28 days post-injury (dpi). The data are expressed as $2^{-\Delta \Delta \mathrm{Ct}}$. (**f**) Confocal microscopy S100A10 cells in the ventral horns (VHs) of the spinal cord, 6–8 mm caudal to the epicenter of injury. Nuclei are stained with DAPI (blue). Scale 100 µm (**a**,**f**). For all (**b**–**e**): Mann–Whitney test. Values are expressed as medians (minimum, maximum, and first and third quartiles (25th and 75th percentiles)), x—mean, n=5 rat/group, * p<0.05; ** p<0.01.

**Figure 4 ijms-25-03772-f004:**
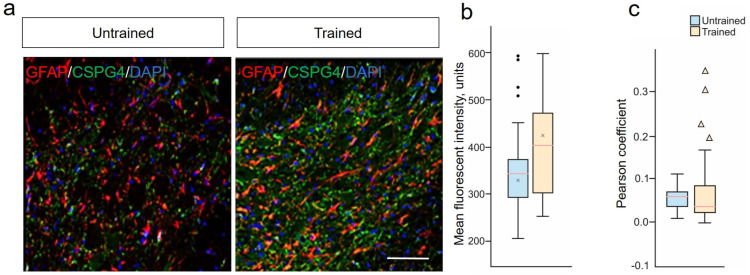
Stability of NG2 glia and limited astrocytic response to treadmill training post-SCI. (**a**) Confocal microscopy of GFAP^+^ (red) and CSPG4^+^ (green) neurons in the ventral horns of the spinal cord, 6–8 mm caudal to the epicenter of injury at 28 dpi. Nuclei are stained with DAPI (blue). Scale 100 µm. (**b**) Quantitative graph of the CSPG4 mean fluorescent intensity. (**c**) Fluorescence colocalization analysis of CSPG4^+^ and GFAP^+^ cells using the Pearson’s coefficient. For all (**b**,**c**): Mann–Whitney test. Values are expressed as medians (minimum, maximum, and first and third quartiles (25th and 75th percentiles)), x—mean, • and ▴—outliers, n=5 rat/group, no significant differences between groups.

**Table 1 ijms-25-03772-t001:** Primary and secondary antibodies for immunofluorescence staining.

Antibody	Host Species	Dilution	Manufacturer
ACAN	Rabbit	1:200	Sigma-Aldrich, St. Louis, MO, USA
ALDH1L1	Rabbit	1:250	Abcam, Cambridge, UK
bTubIII	Mouse	1:100	Abcam
ChAt	Goat	1:100	Abcam
CSPG4	Mouse	1:100	Invitrogen, Waltham, MA, USA
GAP43	Rabbit	1:200	Santa Cruz Biotechnology, Dallas, TX, USA
GFAP	Mouse	1:200	Santa Cruz Biotechnology
NeuN	Rabbit	1:100	Sigma-Aldrich
OPN	Rabbit	1:200	Cloud-Clone Corp., Houston, TX, USA
S100A10	Rabbit	1:100	Abcam
Secondary antibodies (anti-rabbit 488, anti-mouse 555, anti-rabbit 647, anti-goat 647, anti-mouse 488)	-	1:200	Invitrogen

**Table 2 ijms-25-03772-t002:** Nucleotide sequences of primers and probes used for RT-PCR.

Gene Name Primer	Probe Sequences
rGapdh For	ATGACTCTACCCACGGCAAG
rGapdh Rev	TGGAGGATGGTGATGGGTTT
rGfap For	CCAAAGCCTCAAGGAGGAGA
rGfap Rev	CGATGTCCAGGGCTAGCTTA
rChat For	GAGCCAATCGCTGGTATGAC
rChat Rev	CCCTGACGAGCTTCTTGTTG
rSyp For	CTCCTTCCTCCTCTCCCTCT
rSyp Rev	AGCCTCCTCCACTCAGTCTA
rNg2 For	TCCTGGAGAGAGGTGGAAGA
rNg2 Rev	CAAGCCTGTGTTTGTGGTGA

## Data Availability

The data presented in this study are available on request from the corresponding author. The data are not publicly available due to the evolving nature of the project.

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
