# Peer review of "The Impact of Treadmill Training on Tissue Integrity, Axon Growth, and Astrocyte Modulation"

_ijms, 2024, doi:10.3390/ijms25073772_

Round 1
Reviewer 1 Report
Comments and Suggestions for Authors
The study looked at the effects of treadmill training on tissue integrity, axon growth, and astrocytes after a spinal cord. They used well-established methods for the investigation. They found that axon growth was enhanced and a reduction in secondary damage.
Recommendations:
- In the introduction, in lines 49-54, you mention variabilities due to different training methods, but it is unclear how this statement fits with your hypothesis. Please revise the statements.
- Additionally, the purpose of lines 55-58 is unclear. Please revise the statements to fit with the hypothesis.
- Please add the age of the rats when they were euthanized to the methods section.
- Please add how the group averages were determined to the methods. Were repeat measures taken? If so, repeat measures for an animal should have been averaged to create an animal mean, and then all the animal means should have been averaged to create a group means. Please revise all figures to ensure they are correct.
- Please add the item # for the electrodes used and the equipment used to collect the EMG data.
- In the methods section, you list Gfap and Ng2 as primers, but no results are shown. Please add the results or remove those genes from the list of primers.
- In the discussion, you did a great job in describing how your results fit with the rest of the field, but I am unsure what is new or what question you answered.
- In the figures, what do the “x” and “+” symbols represent? They are not defined in the figure legend.
- Additionally, for Figure 1, panels f and f’, there are more than 20 dots. How can that be? Repeat measures for a single animal show be averaged together to create an animal mean.
- In Figure 4, please enlarge panel c to match panel b.
Author Response
Authors: We would like to thank the reviewer for this review and pointing out some errors. We hope that our responses will satisfy the reviewer. In the revised version of the manuscript, all modifications have been highlighted in red font for ease of identification.
The study looked at the effects of treadmill training on tissue integrity, axon growth, and astrocytes after a spinal cord. They used well-established methods for the investigation. They found that axon growth was enhanced and a reduction in secondary damage.
Recommendations:
- In the introduction, in lines 49-54, you mention variabilities due to different training methods, but it is unclear how this statement fits with your hypothesis. Please revise the statements.
- Additionally, the purpose of lines 55-58 is unclear. Please revise the statements to fit with the hypothesis.
Authors: We acknowledge that the initial articulation of our hypothesis, novelty, and objectives may not have been sufficiently clear. We have rephrased the text that was previously in lines 49-54 to more emphatically highlight the novelty of our work, aligning it with our hypothesis.
Previously: “Existing data on molecular and cellular changes in neural tissue observed during motor training in SCI remain contradictory. This variability is due to different training methodologies, including the timing of initiation, duration, and specifics of animal support. In our study, we aimed to mitigate the limited impact of rehabilitation load on the recovery of motor function by focusing on two factors: (1) appropriate selection of SCI severity and (2) enforced plantar placement of hind limbs during treadmill training.”
Now: “Despite the extensive body of research focused on the effects of motor rehabilitation after SCI, existing data on molecular and cellular changes in neural tissue in response to exercise remain contradictory. This variability is largely due to differences in training methodologies, including the timing of initiation, duration, and the specificity of animal support during exercise. There is a noted lack of studies that simultaneously focus on the use of mild spinal cord injury and enforced plantar placement of animal limbs during training sessions. Such an approach could potentially shed light on optimizing conditions for the recovery of motor functions and neuroplasticity, as well as on the mechanisms underlying the efficacy of rehabilitative training.”
With the revised emphasis on novelty, the goal of our study in lines 55-58 appears clearer.
3. Please add the age of the rats when they were euthanized to the methods section.
Authors: We have added in subsections "Immunofluorescent staining" and "RT-qPCR" specific information detailing the time point post-injury at which the animals were euthanized.
Also, in both the previous and current versions of the manuscript, the subsection titled "Animals" specifies the age of the animals at the start of the experiment as "Wistar rats (12-16 weeks old, weighing 200-250 g), all...". Given the initial ages, the animals were effectively 28 days or 4 weeks older at the experiment's end.
4. Please add how the group averages were determined to the methods. Were repeat measures taken? If so, repeat measures for an animal should have been averaged to create an animal mean, and then all the animal means should have been averaged to create a group means. Please revise all figures to ensure they are correct.
Authors: We appreciate your feedback on statistical methodologies. To clarify, we've updated the subsection "Statistical software and data analysis" , specifying that repeat measures, such as different sections from a single rat, were averaged to calculate individual animal means.
We have added "Repeated measurements were averaged to derive a single mean per animal; these means were subsequently combined to determine the overall group average."
This correction has been applied consistently to ensure accurate group averages for each analisys.
For Figure 1 panels f, f’, and h, we've corrected an oversight where data were initially not averaged per animal. The revised figures now correctly represent group means, based on these adjustments.
5. Please add the item # for the electrodes used and the equipment used to collect the EMG data.
Authors: We have updated the "Electromyography" subsection to include the specific item numbers for the electrodes used, as well as details on the equipment employed to collect the EMG data. We utilized an 8-channel electroneuromyograph Neuro-MVP-8 (Neurosoft, Russia) and monopolar needle electrodes with a length of 22 mm and a diameter of 0.40 mm (#MN4022D15SRU, Neurosoft, Russia).
6. In the methods section, you list Gfap and Ng2 as primers, but no results are shown. Please add the results or remove those genes from the list of primers.
Authors: The mRNA analysis results for the genes Gfap and Ng2 are presented in the results section of both the previous and revised versions of the manuscript. For Gfap, "The increase in the number of GFAP+-astrocytes is confirmed by the analysis of mRNA GFAP levels, which showed a more than 3.4-fold increase (p<0.05) at 6-8 mm caudal to the injury site in the Trained group compared to the Untrained group (Fig. 3e)" was performed in the subsection "3.5. Treadmill training promoted anti-inflammatory reactive astrocytes." This is also illustrated in Fig. 3e.
Regarding Ng2 results, we stated, "Also, no significant differences were recorded between groups in the expression of the ng2/cspg4 gene at 6-8 mm caudal to the injury epicenter," in the subsection "3.6. Stability of NG2 glia and limited astrocytic response to treadmill training post-SCI."
7. In the discussion, you did a great job in describing how your results fit with the rest of the field, but I am unsure what is new or what question you answered.
Authors: Given the revised articulation of the novelty of our work, it appears that our discussion of the obtained results and their alignment with other studies may be clearer. Our research stands out for being one of the first to simultaneously investigate the effects of treadmill training on spinal cord recovery post-mild contusion injury, with a specific focus on enforced plantar placement of animal limbs. This dual approach allows us to delve into the molecular and cellular changes within neural tissue.
8. In the figures, what do the “x” and “+” symbols represent? They are not defined in the figure legend.
Authors: The "x" and "+" symbols represent the mean and outliers, respectively, in the graphs of Fig. 1-3, while the "•" (bullet) and "â–²" (triangle) symbols indicate outliers in the graphs of Fig. 4. We have included these definitions in the respective figure legends for clarity.
9. Additionally, for Figure 1, panels f and f’, there are more than 20 dots. How can that be? Repeat measures for a single animal show be averaged together to create an animal mean.
Authors: Each rat in our study yielded no less than 7 sections for analysis, contributing to the large number of measurements per group (n=10 rats per group). We recognize that our initial approach of including all measurements without averaging them for each animal was not in alignment with standard statistical practices for group mean calculation. This oversight has been corrected in the revised figures for panels 1f, f', and h.
10. In Figure 4, please enlarge panel c to match panel b.
Authors: We have adjusted Figure 4 to enlarge panel c to match panel b as requested.

Reviewer 2 Report
Comments and Suggestions for Authors
I thank for the opportunity to review this manuscript. It presents an innovative study about the effects of treadmill training on spinal cord injury recovery, emphasizing motor function enhancement, neural tissue preservation, and axonal growth.
The introduction presents a clarifying rationale of the study.
The objectives are clearly stated.
However, I have some relevant concerns highlighted bellow, related with the clarity in the presentation of the methodology and the lack of conclusions section.
General comments
Please, include the figures in the text of the manuscript after the text in which they are mentioned.
Specific comments
Introduction
Line 46. Please remove the first “signaling”.
Materials and Methods
Line 80: “Body weight support was adjusted based on functional recovery as assessed by the Basso, Beattie, and Bresnahan (BBB) locomotor.” Please, clarify a little, how the weight support was adjusted based on functional recovery.
Line 86. “with scores from 0 to 21…” With better punctuation implying better movement recovery?
LINE 93. Please, add that you also studied: Somatosensory and motor evoked potentials.
Line 123. Please explain the acronym DAPI, it is the first time it appears in the manuscript. I suppose: 4',6-diamidino-2-phenylindole.
Line 125. Please syntax review: To assess areas associated with potential glial scarring glial fibrillary acidic protein/aldehyde dehydrogenase 1 family member L1 (GFAP, ALDH1L1), the following regions of the spinal cord were selected, in line with our previous studies [28; 29].
Line 149. Please, explain the acronyms: dNTPs, and GABDH, it is the first time they appear in the manuscript.
Line 158. Please, add that you have performed some correlations.
Discussion
Line 361. Please syntax review: “al., 2023, demonstrated a significant increase in the number of ChAT+ -motoneurons at the C4 level (rostral to the injury site) [41].”
And clarify, I’m not sure that the results of the study allow to say that neural conductivity was ameliorated with the training.
Conclusions
Please add a conclusion section.
References.
Please, complete the reference list. It starts at reference number 12.
Author Response
Authors: We would like to thank the reviewer for this review and pointing out some errors. We hope that our responses will satisfy the reviewer. We have highlighted all the changes in the manuscript in red for your convenience.
I thank for the opportunity to review this manuscript. It presents an innovative study about the effects of treadmill training on spinal cord injury recovery, emphasizing motor function enhancement, neural tissue preservation, and axonal growth.
The introduction presents a clarifying rationale of the study.
The objectives are clearly stated.
However, I have some relevant concerns highlighted below, related with the clarity in the presentation of the methodology and the lack of conclusions section.
General comments
Please, include the figures in the text of the manuscript after the text in which they are mentioned.
Authors: We want to report that during the layout process in LaTeX, we encountered some difficulties with the precise placement of figures within the text. However, we have taken steps to improve the readability of the article by moving the figures from the end of the manuscript to the "Results" section. This will allow readers to more conveniently review the material. We are confident that the remaining issues regarding figure placement will be successfully resolved by the editorial assistant upon the article's acceptance for publication.
Specific comments
Introduction
Line 46. Please remove the first “signaling”.
Authors: The first instance of "signaling" has been removed and the sentence corrected as requested in line 46.
Materials and Methods
Line 80: “Body weight support was adjusted based on functional recovery as assessed by the Basso, Beattie, and Bresnahan (BBB) locomotor.” Please, clarify a little, how the weight support was adjusted based on functional recovery.
Authors: We have revised this sentence to make it clearer.
Previously: “Body weight support was adjusted based on functional recovery as assessed by the Basso, Beattie, and Bresnahan (BBB) locomotor rating scale.”
Now: “The manual enforcement of weight-supported plantar foot placement on the treadmill were maintained until the animal could perform these actions independently.”
Line 86. “with scores from 0 to 21…” With better punctuation implying better movement recovery?
Authors: We have clarified the sentence for improved understanding.
Previously: “BBB scoring method, with scores from 0 to 21, was employed to evaluate voluntary 86 movement recovery”
Now: “The Basso, Beattie, and Bresnahan (BBB) scoring method, which employs a scale ranging from 0 to 21, was used to assess the recovery of voluntary movement, where higher scores indicate better movement recovery”
LINE 93. Please, add that you also studied: Somatosensory and motor evoked potentials.
We added it “Somatosensory (SEPs) and motor (MEPs) evoked potentials, M- and H-waves…”
Line 123. Please explain the acronym DAPI, it is the first time it appears in the manuscript. I suppose: 4',6-diamidino-2-phenylindole.
Authors: We have included the full name of DAPI, 4',6-diamidino-2-phenylindole, at its first mention in the manuscript to ensure clarity. “...with cell nuclei counterstained using 4’,6-diamidino-2-phenylindole (DAPI).”
Line 125. Please syntax review: To assess areas associated with potential glial scarring glial fibrillary acidic protein/aldehyde dehydrogenase 1 family member L1 (GFAP, ALDH1L1), the following regions of the spinal cord were selected, in line with our previous studies [28; 29].
Previously: “To assess areas associated with potential glial scarring glial fibrillary acidic protein/aldehyde dehydrogenase 1 family member L1 (GFAP, ALDH1L1), the following regions of the spinal cord were selected, in line with our previous studies…”
Now: “To assess areas associated with potential glial scarring glial fibrillary acidic protein (GFAP) and aldehyde dehydrogenase 1 family member L1 (ALDH1L1), the following regions of the spinal cord were selected, in line with our previous studies…”
Line 149. Please, explain the acronyms: dNTPs, and GABDH, it is the first time they appear in the manuscript.
Authors: We have included the full expansion of the acronym at its first mention in the text:
“deoxynucleotide triphosphates (dNTPs)”. We have corrected the typo. Indeed, the accurate acronym for glyceraldehyde 3-phosphate dehydrogenase is GAPDH, not GABDH.
Line 158. Please, add that you have performed some correlations.
Authors: We have added: “The Pearson correlation was carried out to quantify the correlation between BBB scores and histological data of pathological cavities.” in subsections "Statistical software and data analysis methods".
Discussion
Line 361. Please syntax review: “al., 2023, demonstrated a significant increase in the number of ChAT+ -motoneurons at the C4 level (rostral to the injury site) [41].”
And clarify, I’m not sure that the results of the study allow to say that neural conductivity was ameliorated with the training.
Authors: We acknowledge that our results regarding the absence of a significant effect on a specific subpopulation of neurons, despite analysis indicating improvements in neurological assessments, might appear paradoxical, particularly when compared to other studies. In the discussion of our results, we emphasize this point: "The lack of effect of treadmill training on the subpopulation of ChAT+, OPN+, and PARV+ motoneurons and the expression of the chat gene in our study might seem counterintuitive, especially in the context of results from other studies."
This discrepancy underscores a complex aspect of SCI recovery that our findings alone cannot fully elucidate. It suggests that while treadmill training may lead to general improvements in neural conductivity, as inferred from overall neurological assessments, its direct impact on specific neuronal subpopulations and molecular markers may differ and warrants further exploration.
Our observations imply that the mechanisms underpinning improvements in neural conduction and functional recovery post-SCI may extend beyond the modulation of the phenotypes and expressions of the neurons and markers we examined. These results highlight the need for a deeper and more comprehensive understanding of the multifaceted nature of SCI recovery. This includes investigating the roles of other neuronal populations, interneuronal connections, and molecular pathways not addressed in our current research. Furthermore, it necessitates future studies to explore other areas of the injured spinal cord, particularly regions rostral to the injury site as done in Lin et al., 2023, or to focus on in the lumbar section of the injured spinal cord.
Conclusions
Please add a conclusion section.
Authors: The revised and previous (page 10 of PDF document) versions of the manuscript include a conclusion section.
References.
Please, complete the reference list. It starts at reference number 12.
Authors: The revised and previous versions of the manuscript include all references.

Round 2
Reviewer 1 Report
Comments and Suggestions for Authors
Thank you for addressing most of my comments. However, Figure 4 panel C still does not match panel b. I think you should make panel c look similar to panel b.
Author Response
Dear Reviewer,
Thank you for your attentive review and for highlighting the discrepancy in Figure 4, panel C. We regret that we initially overlooked this error in the PDF file. Following your suggestion, we have now corrected Figure 4 to ensure that panel C matches panel B. We appreciate your guidance in improving the quality of our manuscript and thank you for your patience and understanding.
Best regards,
Ageeva Tatyana

Reviewer 2 Report
Comments and Suggestions for Authors
I want to thank the authors for the implementation of all the corrections proposed. I apologize because of the corrections proposed in the round 1, on the lack of conclusions and the first numbers of the reference list. It has been a problem with my adobe application.
Congratulations for the manuscript.
Author Response
Dear Reviewer,
We deeply appreciate your kind words and acknowledgment regarding the implementation of the corrections proposed in round 1.
Please accept our sincere thanks for your constructive feedback throughout the review process, which has undeniably contributed to enhancing the quality of our manuscript. Your expertise and thoughtful recommendations have been invaluable.
Best regards,
Ageeva Tatyana